# Effects of Dietary Metabolizable Energy and Crude Protein Levels on the Nutrient Metabolism, Gut Development and Microbiota Composition in Jingyuan Chicken

**DOI:** 10.3390/ani15162387

**Published:** 2025-08-14

**Authors:** Xin Guo, Jie Liu, Jie Yang, Qiaoxian Gao, Juan Zhang, Wenzhi Yang, Guosheng Xin

**Affiliations:** 1College of Life Sciences, Ningxia University, Yinchuan 750021, China; gu0xin234@163.com (X.G.); liujiefrom1999@126.com (J.L.); yangjienxu@163.com (J.Y.); gaoqx@nxu.edu.cn (Q.G.); yangwz22@nxu.edu.cn (W.Y.); 2Ningxia Feed Engineering Technology Research Center, Ningxia University, Yinchuan 750021, China; 3School of Animal Science and Technology, Ningxia University, Yinchuan 750021, China; zhangjuannxy@nxu.edu.cn; 4Key Lab of Ministry of Education for Protection and Utilization of Special Biological Resources in Western China, Ningxia University, Yinchuan 750021, China

**Keywords:** crude protein, energy, growing chicken, ileum histomorphometry, gut microbiota

## Abstract

Jingyuan chicken is a distinctive local breed known for its tolerance to low-quality feed and robust environmental adaptability. However, traditional feeding standards fail to meet its precise nutritional requirements. Therefore, this study systematically optimized dietary energy and protein levels and investigated their regulatory mechanisms governing growth. Results demonstrated that a medium energy (11.70 MJ/kg) and protein (15.5%) diet improved intestinal development and microbiota composition, significantly enhancing growth performance. These findings provide a practical and efficient feeding regimen for Jingyuan chicken farming.

## 1. Introduction

Metabolizable energy (ME) and crude protein (CP) are primary factors influencing the nutritional quality of poultry diets. These factors not only directly affect the growth performance of livestock and poultry but also play a crucial role in intestinal development, microbial community composition, and nutrient absorption efficiency [1,2]. Excessive or insufficient energy and protein intake typically reduces intestinal villus height, restricts intestinal development, and lowers the rate of metabolism of nutrients, consequently retarding growth and weight gain in chickens [3,4]. Additionally, inadequate dietary energy restricts the efficient utilization of dietary proteins, leading to reduced protein metabolism efficiency and increased ammonia and urea emissions [5]. These outcomes extend beyond merely affecting intestinal morphology and nutrient metabolism and further include changes in gut microbiota composition [6]. When chickens are fed diets characterized by imbalanced energy and protein levels, pathogenic bacteria proliferate and beneficial microbial populations are suppressed, resulting in gut dysbiosis and metabolic disorders [7]. Shifts in gut microbiota composition are closely linked to the intestinal health and growth performance of chickens [8]. Appropriate energy and protein levels are essential in enhancing the abundance of beneficial bacteria, such as *Bifidobacteriaceae, Paraprevotella*, and *Lactobacillus crispatus*; promoting intestinal development; and improving the efficiency of nutrient absorption [9,10]. Thus, the gut microbiota serves as a key mediator in the optimal utilization of the effects of dietary energy and protein on chicken growth and development [11,12].

Indigenous breeds (e.g., Jingyuan chicken) exhibit unique digestive characteristics compared with commercial poultry breeds, particularly with respect to their tolerance for coarse feed and adaptability to energy and protein requirements [13]. Studies suggest that indigenous chickens have relatively lower energy and protein demands, making appropriately reduced dietary energy and protein levels more conducive to leveraging their genetic traits [7,14]. For instance, yellow feathered chickens demonstrate distinct nutritional needs, requiring lower energy and protein levels compared with white-feathered chickens (commercial breeds), enabling them to sustain growth and maintain strong stress resistance under comparatively lower energy and protein conditions [1]. In contrast, there are very few studies on the energy and protein requirements of Jingyuan chicken, a high-quality indigenous breed in Ningxia. Notably, under the local rearing system, Jingyuan chickens are raised in a centralized facility during the brooding phase and then transferred to individual farmers for the growing period from 7 to 18 weeks of age. Despite the importance of this phase for rapid growth and gastrointestinal development, a standardized dietary formulation tailored to this specific stage is still lacking. At 18 weeks of age, Jingyuan chickens reach the onset of sexual maturity, by which time the digestive system has generally completed its structural and functional development [15]. Therefore, clarifying the nutritional requirements and growth performance of Jingyuan chickens during the growing period under varying energy and protein levels is essential for precisely regulating their feed and diet.

An optimal energy and protein ratio maximizes broiler growth while avoiding adverse effects on production performance and health resulting from nutrient deficiency or metabolic overload [16]. While studies on Jingyuan chickens have focused on genetic and germplasm resources, studies on nutritional requirements remain limited, posing a challenge in establishing feed standards and precise nutritional management for efficient farming. Thus, there is a pressing need to study optimal energy and protein regulation in advancing Jingyuan chicken production. The aim of this study was to investigate the effects of diets with varying energy and protein levels on the nutrient metabolism, digestive organ indices, intestinal morphology, and microbial structure of male Jingyuan chickens aged 7–18 weeks. Our findings will help lay both theoretical and practical foundations in formulating nutritional requirements and standards during the growth period of meat-type Jingyuan chickens.

## 2. Materials and Methods

### 2.1. Experimental Design and Diets

A total of 540 seven-week-old healthy male Jingyuan chickens (418.19 ± 1.25 g) with uniform body weights were obtained from Ningxia Wansheng Industrial Co. Ltd. (Ningxia, China) and randomly divided into 9 groups, with 6 replicates per group and 10 chickens per replicate. Each replicate comprised 5 cages, with 2 chickens per cage. A 3 × 3 factorial experimental design was adopted, with three dietary ME levels, namely, 11.28 MJ/kg (low ME, LE), 11.70 MJ/kg (medium ME, ME), and 12.12 MJ/kg (high ME, HE), and three dietary CP levels, namely, 14.00% (low CP, LP), 15.50% (medium CP, MP), and 17.00% (high CP, HP). The following nine diets were formulated in total: low ME and low CP (LELP) group, low ME and medium CP (LEMP) group, low ME and high CP (LEHP) group, medium ME and low CP (MELP) group, medium ME and medium CP (MEMP) group, medium ME and high CP (MEHP) group, high ME and low CP (HELP) group, high ME and medium CP (HEMP) group, and high ME and high CP (HEHP) group. ME and CP levels were primarily based on China’s Feeding Standard of Chickens (NY/T 33—2004) [17], with appropriate adjustments made to the ME and CP settings. The composition and nutritional levels of the experimental diets are detailed in Table 1.

### 2.2. Feeding Management

Prior to the experiment, a comprehensive cleaning and sterilization protocol was implemented to clean the chicken cages and associated equipment. The experiment was conducted in a closed, environmentally controlled poultry house. A three-tier semi-stepped cage system was used, with two chickens per cage and even distribution across tiers. Dimensions of each cage were 40 cm × 50 cm × 50 cm in width, length, and height. The environmental conditions were controlled at 20–25 °C, 50–60% relative humidity, and 16 h of light per day at 20 lux. Chickens underwent a 1-week adaptation period before the 10-week formal trial. Daily management included ventilation, cleaning, and routine health checks.

### 2.3. Sample Collection

The initial and final weights of Jingyuan chickens were recorded at the commencement and conclusion of the formal trial, respectively. During the trial, the feed intake was recorded on a weekly basis, and the average daily feed intake (ADFI), average daily gain (ADG), and feed conversion ratio (FCR) were calculated. The number of deceased Jingyuan chickens was recorded on a daily basis, and the FCR was corrected based on the mortality data.

One week before the end of the trial, a metabolism experiment was conducted using the total fecal-collection method. Feed intake and excreta weight were accurately recorded for each replicate over 1 week, and contemporaneous feed samples were collected for subsequent nutrient analysis. Excreta samples collected in trays were inspected daily. Foreign matter (e.g., feathers) was removed and the samples were sprayed with an appropriate amount of 10% concentrated sulfuric acid for nitrogen fixation. The weights of fresh excreta samples were recorded. Subsequently, fresh excreta and feed samples were dried, weighed, ground, sieved, and stored for subsequent chemical analysis [19].

After 12 h of fasting at the end of the formal trial, six male chickens from each group with body weights close to the average body weight of the group were randomly selected for slaughter. Chickens were anesthetized by inhalation of ether until loss of consciousness, followed by humane euthanasia via severing of the carotid artery. The glandular stomach and gizzard were emptied of their contents and weighed, and the lengths of the duodenum, jejunoileum, and cecum were measured [20].

A 3–5-cm segment of jejunal tissue (located 5–10 cm below the junction of the duodenum and the jejunum) was excised, gently flushed with physiological saline to remove chyme, and fixed in 4% formaldehyde solution for 24 h for morphological analysis. Cecal contents were collected and snap-frozen in liquid nitrogen for microbiota sequencing.

### 2.4. Apparent Nutrient Metabolism

DM, GE, and CP contents were determined according to GB/T 6435-2014 [21], GB/T 45104-2024 [22], and GB/T 6432-1994 [23], respectively. Fresh excreta and feed samples were dried at 105 °C for 24 h, and the DM content was calculated based on the weight after drying. GE was measured using an adiabatic bomb calorimeter (C 5001, IKA, Stuttgart, Germany) and the results are expressed in MJ/kg. CP was analyzed using the Kjeldahl nitrogen method, and nitrogen content was determined using a nitrogen analyzer (KDN103F, Shanghai Fiber Inspection Instrument Co., Ltd., Shanghai, China). CP content was calculated as 6.25 × Kjeldahl nitrogen.

### 2.5. Jejunal Morphology

Following sample collection, tissues were dehydrated through a graded ethanol series and cleared in xylene, then embedded in paraffin. The trimmed paraffin blocks were sectioned into 4-μm-thick slices using a microtome and stained with hematoxylin and eosin [24]. Ten well-oriented villus and crypt regions were selected for analysis. Villus height (VH), crypt depth (CD), and muscle layer thickness (MLT) were determined using a light microscope (NIKON ECLIPSE E100, NIKON, Tokyo, Japan) and Image-Pro Plus 6.0 software (Media Cybernetics, Inc., Rockville, MD, USA).

### 2.6. 16S rDNA Sequencing and Analysis

Based on growth performance and intestinal development, the MEMP group exhibited the best results. To compare the effects of different energy levels at the medium protein level, the LEMP, MEMP, and HEMP groups were selected. Similarly, to assess the impact of varying protein levels at a medium energy level, the MELP, MEMP, and MEHP groups were chosen. At the end of the 77-day feeding trial, the cecal contents were collected for 16S rDNA sequencing analysis. Total genomic DNA from cecal contents was extracted using a soil and fecal genomic DNA extraction kit, following the manufacturer’s instructions, and stored at −20 °C until further use. The V3–V4 region of the bacterial 16S rDNA gene was amplified using PCR using the forward primer 341F (5′-CCTAYGGGRBGCASCAG-3′) and reverse primer 806R (5′-GGACTACNNGGGTATCTAAT-3′). The PCR products were purified using 2% agarose gel electrophoresis and recovered using a DNA purification kit. Sequencing was performed on the Illumina NovaSeq 6000 platform.

### 2.7. Data Processing and Statistical Analysis

Statistical analyses were performed using SPSS 23.0. After confirming that the data were normally distributed, variance was analyzed using the General Linear Model. Differences between groups were compared using Duncan’s multiple range test. Differences were considered significant at *p* < 0.05.

## 3. Results

### 3.1. Growth Performance

As shown in Table 2, varying levels of ME and CP had significant effects on ADG and FCR, with a notable interaction between the two factors (*p* < 0.001). While ME and CP levels did not affect the ADFI, their interaction exhibited significant differences. The combination of 11.70 MJ/kg ME and 15.5% CP resulted in the highest ADG and the lowest FCR among the groups (*p* < 0.05). When the dietary CP level was 17%, the group receiving 12.12 MJ/kg ME exhibited a significantly higher ADFI than the other ME groups (*p* < 0.05). At dietary ME levels of 11.70 or 12.12 MJ/kg, the 17.0% CP group showed a significantly lower FCR than the 14.0% CP groups (*p* < 0.05).

### 3.2. Apparent Nutrient Metabolism

As shown in Table 3, different levels of ME and CP significantly affected the apparent metabolic rates of CP, with a notable interaction between the two factors (*p* < 0.05). Dietary ME level significantly influenced the apparent metabolizable energy of GE, while CP level had no noticeable impact (*p* > 0.05). Meanwhile, neither ME nor CP levels significantly impacted the apparent metabolic rate of DM (*p* > 0.05). At an ME level of 11.28 MJ/kg, the apparent metabolic rate for CP in the 17% CP group was significantly lower than in the other CP groups (*p* < 0.05) while, at CP levels of 14.0%, the apparent metabolic rate of CP in the 12.12 MJ/kg ME group was significantly lower compared to the other ME groups (*p* < 0.05).

### 3.3. Digestive Organ Index

It can be seen in Table 4 that varying levels of ME significantly affected the weights of the proventriculus and gizzard and the lengths of the duodenum and cecum (*p* < 0.05). In contrast, varying CP levels led to significant changes in the lengths of the jejunoileum and cecum (*p* < 0.05). A significant interaction between ME and CP levels (*p* < 0.05) was observed with respect to gizzard weight and duodenum length. Specifically, when the CP level was 17%, the group of chickens receiving 12.12 MJ/kg ME exhibited significantly higher gizzard weights and duodenum length compared with chickens in the other ME groups (*p* < 0.05). When CP was 15.5%, the duodenum length in chickens in the 11.70 MJ/kg ME group was significantly higher than that in the other CP groups of chickens (*p* < 0.05).

### 3.4. Jejunal Morphology

As seen in Table 5, ME levels significantly influenced VH, MLT, and VH/CD, whereas CP levels significantly affected CD, VH, and MLT (*p* < 0.05). A significant interaction between ME and CP levels was observed for the morphological parameters in the jejunum, including CD and VH (*p* < 0.01). Specifically, when the CP level was 15.5%, the group receiving 12.12 MJ/kg ME exhibited significantly lower CD, VH, VH/CD and MLT values (*p* < 0.05) compared with those in the other ME groups, along with shorter villi and thinner mucosa (Figure 1 HEMP). Conversely, when the ME level was 11.70 MJ/kg, the 15.5% CP group showed significantly higher VH, VH/CD, and MLT values (*p* < 0.05) compared with those in the other CP groups, and the jejunal villi appeared more elongated and closely arranged (Figure 1 MEMP). However, CD did not differ significantly among the CP groups (*p* > 0.05).

### 3.5. 16S rDNA Analysis of Cecal Microbiota

The intestinal microbiota of Jingyuan chickens in the LEMP, MELP, MEMP, MEHP, and HEMP groups shared 760 operational taxonomic units (OTUs), as shown in Figure 2A. When the relative abundance of OTUs in each group dropped below 10^−4^, the rarefaction curves gradually leveled off, indicating sufficient sequencing depth and a relatively even species distribution, as illustrated in Figure 2B. Figure 2C shows that no significant differences (*p* > 0.05) in the Shannon, Simpson, and Chao 1 indices were noted among the groups with respect to cecal microbiota.

The relative abundance of cecal microbiota in Jingyuan chickens was analyzed at both the phylum and genus levels, as presented in Figure 3 and Figure 4. At the phylum level, Bacteroidota, Firmicutes, Proteobacteria, and Desulfobacterota were the dominant taxa in the cecum, as shown in Figure 3A. The relative abundance of Desulfobacterota was significantly higher in the MELP group (2.91% vs. 4.45%) versus that in the MEMP group. Compared with the MELP group, the MEHP group exhibited a significant decrease in the abundance of Desulfobacterota (4.45% vs. 2.45%) and Synergistota (0.79% vs. 0.48%). Additionally, the LEMP group showed a significant increase in the abundance of Fusobacteriota (0.75% vs. 2.0%) and a decrease in the abundance of Synergistota (0.79% vs. 0.41%) relative to the MELP group, as demonstrated in Figure 4A.

*Bacteroides*, *Romboutsia*, *Lactobacillus*, *Rikenellaceae_RC9_gut_group*, and *Muribaculaceae* were identified as the dominant genera in the cecum. The relative abundance of *Clostridia_vadinBB60_group* was significantly lower in the MEHP group than that in the MEMP group (0.54% vs. 0.79%), whereas the relative abundance of *F082* was significantly higher in both the MELP (0.24% vs. 1.62%) and LEMP (0.24% vs. 0.84%) groups. The MEHP group exhibited a significant decrease in *Rikenellaceae_RC9_gut_group* (8.60% vs. 4.27%), *Clostridia_vadinBB60_group* (0.81% vs. 0.54%), and *Olsenella* (0.27% vs. 0.19%) compared with that in the MELP group. Relative to the LEMP group, the MEHP group showed a significant decrease in the abundance of *Odoribacter* (0.62% vs. 0.30%). The MELP group had a significantly lower relative abundance of *Bacteroides* (34.56% vs. 27.95%), *Fusobacterium* (2.0% vs. 0.75%), and *Odoribacter* (0.62% vs. 0.29%) but a higher relative abundance of *Rikenellaceae_RC9_gut_group* (5.70% vs. 8.60%), *Synergistes* (0.41% vs. 0.78%), and *Escherichia-Shigella* (0.22% vs. 0.49%) compared with that in the LEMP group. The HEMP group exhibited a significant reduction in the relative abundance of *F082* (0.84% vs. 0.32%) and *Parabacteroides* (1.36% vs. 0.94%) but an increase in the relative abundance *Megasphaera* (0.53% vs. 1.06%) compared with that in the LEMP group. These differences are illustrated in Figure 4B.

Figure 5 presents the results of linear discriminant analysis; effect size was used to identify the bacterial taxa in the cecal samples of Jingyuan chickens provided different levels of ME and CP. Erysipelotrichaceae, Syntrophomonadaceae, and *Akkermansia* were notably enriched in the MEMP group, whereas *F082*, *Odoribacter,* Rikenellaceae, Desulfobulbaceae, and *Clostridia_vadinBB60_group* were enriched in the LEMP and MELP groups. *Ruminococcus* was predominantly enriched in the HEMP and MEHP groups.

## 4. Discussion

### 4.1. Growth Performance and Apparent Nutrient Metabolism

Energy and protein levels are fundamental components of livestock and poultry diets, with their composition and balance playing a crucial role in nutrient utilization and production performance [25]. The results of this study demonstrate that dietary metabolizable energy (ME) and crude protein (CP) levels have a significant interactive effect on the growth performance of Jingyuan chickens, consistent with the observations of Zhu et al. [26]. Specifically, when the ME level was 11.28 MJ/kg and CP was 14%, the average daily gain (ADG) of Jingyuan chickens significantly decreased, indicating that a low-energy and low-protein combination could reduce feed palatability and intake, thereby suppressing growth rate [27]. Although chickens fed low-protein diets exhibit reduced ADG, they demonstrate improved nitrogen utilization efficiency [28], a trend that was consistent with the findings of our study. The study found that feed intake decreased significantly in Jingyuan chickens with increasing CP levels, while elevated ME levels promoted feed intake. Ko et al. reported that feed intake significantly decreased in male Cobb broilers when the protein level was 19% and dietary energy decreased by 100 kcal/kg, indicating a correlation between dietary energy level and feed intake [3]. However, previous studies have indicated that feed intake and FCR of Beijing-You chickens and Taihe silky fowls were significantly affected by dietary ME levels, and increasing ME levels would lead to reductions in both feed intake and FCR in these breeds [7,9]. In addition, this study observed a significant interaction between ME and CP levels on Jingyuan chicken FCR. In contrast, increasing dietary CP resulted in higher FCR in male Arbor Acres broilers, while ME levels had no significant effect [5]. This demonstrates breed-specific differences in the nutritional responses to dietary energy and protein. The findings of our study confirm that inappropriate dietary energy and protein levels—whether insufficient, excessive, or imbalanced—can negatively affect the growth performance and nutrient metabolism of Jingyuan chickens. Therefore, optimizing the ME-to-CP ratio is crucial in improving feed efficiency and enhancing nutrient digestion and absorption in chickens [29]. Our findings indicate that male Jingyuan chickens in the MEMP group (11.70 MJ/kg ME, 15.50% CP) showed higher ADG and feed efficiency during the growing period.

However, this study has certain limitations. The dietary recommendation derived from the MEMP group is applicable only to male Jingyuan chickens, and its relevance to females remains untested. In addition, the experimental period was limited to 7–18 weeks, without evaluating the early brooding phase or other production stages. Further research covering other developmental periods is needed to support recommendations for the entire Jingyuan chicken production. Nevertheless, the findings are valuable for the indigenous Jingyuan chicken industry. By identifying the optimal combination of dietary energy and protein levels, this study contributes to the development of tailored feeding strategies for Jingyuan chickens, which can improve production efficiency and gut health, while also supporting the development of regional poultry farming.

### 4.2. Digestive Organ Index and Jejunal Morphology

The proventriculus and gizzard are the primary digestive organs responsible for the grinding of food and secretion of gastric acid in chickens. Their development is influenced by dietary ME and CP levels [30,31]. In our study, increasing dietary ME levels while maintaining CP levels at 17% resulted in an increase in gizzard weights. Our findings aligns with those reported previously [32], wherein significant interactive effects between ME and CP levels on the gizzard weight in pigeons were noted. Similarly, chickens that were fed diets containing 2550 kcal/kg ME and 17% CP had significantly higher gizzard weights than those fed diets containing 3000 kcal/kg ME and 20% CP [16]. These results suggest that lower energy-to-protein ratios may enhance the mechanical digestion capacity of the gizzard and promote its development.

The duodenum and jejunum are intestinal segments that play a crucial role in nutrient absorption in chickens. Restricted dietary ME and CP levels are known to significantly reduce duodenal and jejunoileal lengths, as well as VH and the absorptive surface area [33,34]. VH exhibits a quadratic response to dietary protein levels, indicating that both very high and very low nutrient densities may impair intestinal development [4,35]. Furthermore, an imbalanced ME-to-CP ratio can decrease nutrient metabolism efficiency, interfere with intestinal cell proliferation (e.g., mitosis of jejunal cells), and hinder intestinal development [32,36]. Optimizing the dietary ME-to-CP ratio can significantly improve intestinal structure and function. For example, adjusting ME and CP levels can enhance the VH in the duodenum and ileum, increase the villus height-to-crypt depth (VH/CD) ratio, and optimize intestinal morphology [4,37]. In the current study, chickens subjected to a high daily-gain diet (MEMP group, 11.70 MJ/kg ME and 15.50% CP) exhibited significantly higher duodenal length, jejunal VH, and muscularis thickness. Additionally, the interaction between ME and CP levels significantly altered these indices. Overall, an appropriate ME-to-CP ratio can contribute to the optimization of intestinal and villus structures, promoting nutrient absorption.

### 4.3. Cecal Microbiome

Bacteroidota, Firmicutes, and Desulfobacterota are the dominant microbial phyla in the cecum of chickens that play crucial roles in gut health and energy metabolism. Among them, Bacteroidota and Firmicutes are particularly effective in degrading dietary proteins and carbohydrates, providing essential energy for chickens, facilitating weight gain, and maintaining intestinal homeostasis [38]. Firmicutes can also degrade tryptophan to produce indole compounds, which help prevent intestinal tissue damage and pathogen invasion, thereby maintaining the gut barrier function [39,40]. However, in our study, an increase in the relative abundance of Desulfobacterota was noted in the ceca of chickens fed a low-protein diet, which is consistent with that reported in a previous study [6]. Under low-CP conditions, the proliferation of Desulfovibrio leads to the degradation of amino acids and other carbon substrates to hydrogen sulfide, a compound known to be toxic to enterocytes. Consequently, this process damages the intestinal mucosa and disrupts gut homeostasis [41,42]. A low-protein diet induced the proliferation of *Desulfovibrio* spp., impaired the intestinal mucosa and microenvironment, and reduced the relative abundances of beneficial bacteria, including *Bifidobacterium* and *Clostridium butyricum* [43].

Organic acids are key metabolites that are produced in the gut during microbial fermentation. These acids play a crucial role in supporting the growth and intestinal development of broilers [44]. In the MEMP group, the bacterial families Erysipelotrichaceae, Syntrophomonadaceae, *Akkermansia*, and *Clostridia_vadinBB60_group* were enriched, and these bacteria are associated with the metabolism of acetic acid and propionic acid. Short-chain fatty acids, including acetate and propionate, help maintain intestinal barrier function and regulate gut immune responses, thereby promoting the growth and health of chickens [45,46,47]. Although broiler growth performance is suppressed when fed diets low in ME and CP, certain microbial groups, such as *F082* and *Rikenellaceae_RC9_gut_group*, can still improve gut health and promote the growth and production performance via propionate fermentation [48,49]. For example, feeding organic acids can improve gut morphology, effectively increasing crypt depth and the VH/CD ratio and enhancing nutrient absorption [50]. Erysipelotrichaceae and *Akkermansia* are closely involved in organic acid production and play crucial roles in regulating intestinal immunity and inflammation [51]. The abundance of *Odoribacter* in the cecum was also significantly enriched in the LEMP group. *Odoribacter* from the Bacteroidota phylum inhibits the growth of harmful bacteria by modulating fatty acid metabolism and producing short-chain fatty acids, effectively enhancing nutrient absorption [52,53]. Based on these findings, Erysipelotrichaceae, Syntrophomonadaceae, *Akkermansia*, *Clostridia_vadinBB60_group*, and *Odoribacter* may represent a group of bacteria that are potentially beneficial for broiler growth.

## 5. Conclusions

Varying dietary energy and protein levels significantly affected the growth performance, apparent nutrient metabolism, digestive organ traits and gut microbiota composition of growing Jingyuan chickens, and a significant interaction between the two factors was noted. Chickens in the medium energy (11.70 MJ/kg) and medium protein (15.50%) group exhibited the best growth performance and intestinal development. Dietary energy and appropriate protein levels could markedly optimize the gut microbiota structure by suppressing the relative abundance of Desulfobacterota and increasing the abundances of beneficial bacteria, including Erysipelotrichaceae, Syntrophomonadaceae, *Akkermansia, the Clostridia_vadinBB60_group*, and *Odoribacter*, thereby improving the intestinal environment and promoting nutrient absorption and utilization.

## Figures and Tables

**Figure 1 animals-15-02387-f001:**
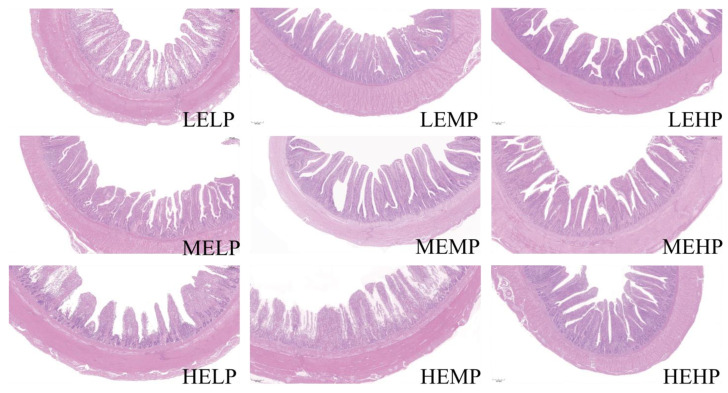
Microscopic images of the jejunum (scale bar = 200 μm). Low ME, low CP group (LELP); low ME, medium CP group (LEMP); low ME, high CP group (LEHP); medium ME, low CP group (MELP); medium ME, medium CP group (MEMP); medium ME, high CP group (MEHP); high ME, low CP group (HELP); high ME, medium CP group (HEMP); high ME, high CP group (HEHP).

**Figure 2 animals-15-02387-f002:**
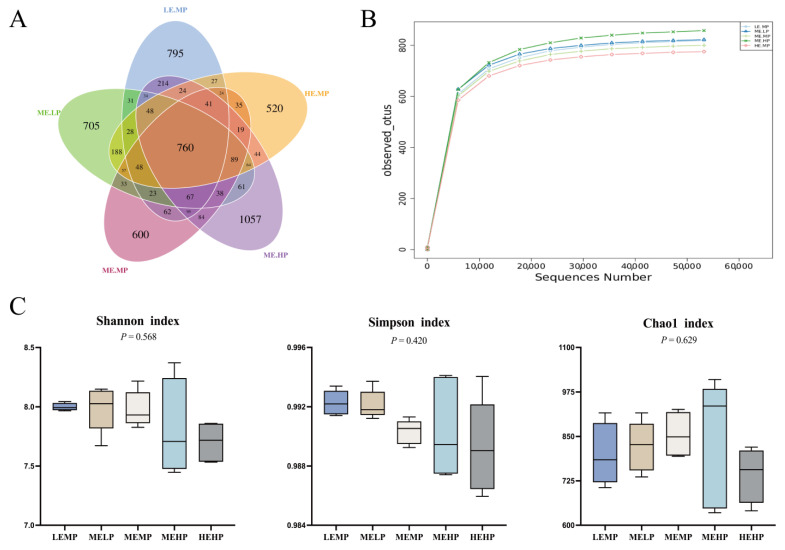
Effects of different ME and CP levels on the α-diversity of cecal microbiota in Jingyuan chickens. (**A**) Venn diagram of cecal microbiota OTUs in the LEMP, MELP, MEMP, MEHP, and HEMP groups. (**B**) Rarefaction curves. (**C**) Shannon and Simpson indices representing microbial community diversity; Chao 1 index representing microbial community richness.

**Figure 3 animals-15-02387-f003:**
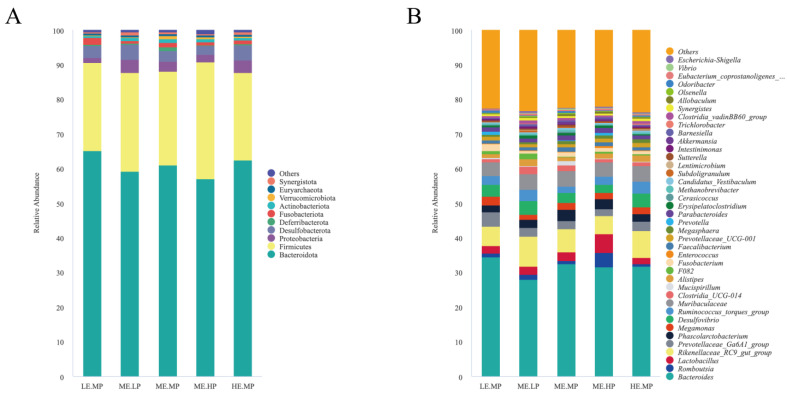
Effects of different ME and CP levels on the composition of cecal microbiota at the phylum and genus levels in Jingyuan chickens. (**A**) Relative abundance of dominant bacterial phyla in each group. (**B**) Relative abundance of dominant bacterial genera in each group. Groups: LEMP (low ME and medium CP), MELP (medium ME and low CP), MEMP (medium ME and medium CP), MEHP (medium ME and high CP), and HEMP (high ME and medium CP).

**Figure 4 animals-15-02387-f004:**
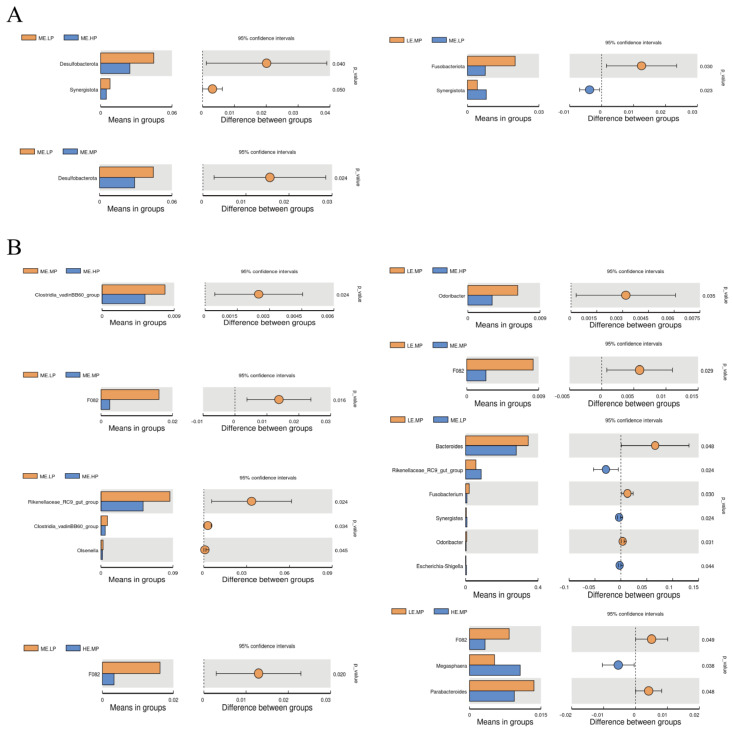
Differential cecal microbiota at the phylum and genus levels in Jingyuan chickens under different ME and CP levels. (**A**) Differential bacterial phyla. (**B**) Differential bacterial genera. Groups: LEMP (low ME and medium CP), MELP (medium ME and low CP), MEMP (medium ME and medium CP), MEHP (medium ME and high CP), and HEMP (high ME and medium CP). Method: T-test.

**Figure 5 animals-15-02387-f005:**
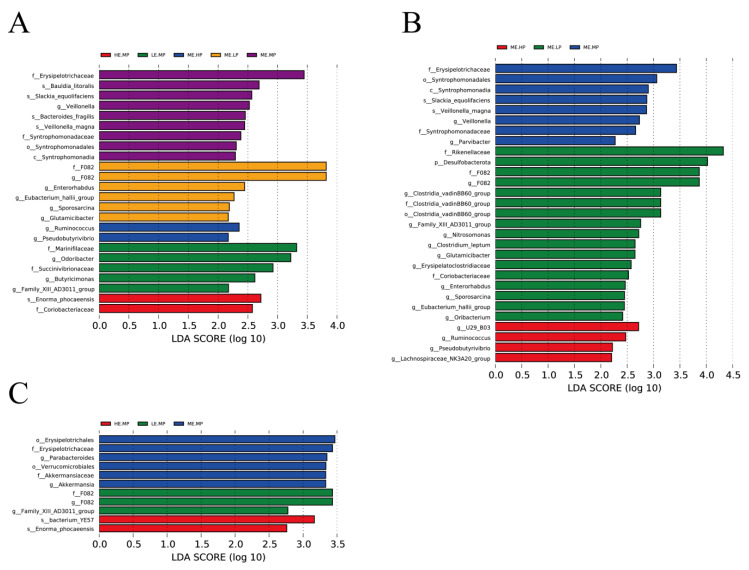
Identification of significantly different taxa among groups using linear discriminant analysis effect size (LEfSe) with default parameters (LDA score = 2). (**A**) Taxa representing significant differences among LEMP (low ME and medium CP), MELP (medium ME and low CP), MEMP (medium ME and medium CP), MEHP (medium ME and high CP), and HEMP (high ME and medium CP) groups. (**B**) Taxa representing significant differences among MELP, MEMP, and MEHP groups. (**C**) Taxa representing significant differences among LEMP, MEMP, and HEMP groups.

**Table 1 animals-15-02387-t001:** Composition and nutrient levels of the basal diet (air-dry basis).

Items	LE			ME			HE		
	LP	MP	HP	LP	MP	HP	LP	MP	HP
Ingredients, %									
Corn	47.32	44.00	47.18	47.92	49.02	46.50	50.27	48.40	47.34
Soybean meal	6.98	11.85	18.10	7.90	13.56	18.78	8.03	12.22	16.10
Wheat middling	16.70	18.00	11.50	18.00	13.80	14.20	14.50	13.70	12.00
Soybean oil	1.00	1.00	1.00	2.00	2.00	2.00	3.40	3.40	3.40
Wheat barn	21.80	19.17	16.60	18.20	15.80	12.90	16.94	14.94	12.90
Corn protein meal	1.10	1.16	1.00	1.00	1.00	1.00	1.82	2.45	3.50
Mountain flour	1.72	1.72	1.66	1.64	1.60	1.58	1.60	1.58	1.54
CaHPO_4_	1.30	1.26	1.29	1.38	1.40	1.38	1.45	1.44	1.45
NaCl	0.50	0.40	0.40	0.40	0.40	0.40	0.40	0.40	0.40
Premix ^1^	1.00	1.00	1.00	1.00	1.00	1.00	1.00	1.00	1.00
Lys	0.34	0.23	0.09	0.33	0.22	0.08	0.35	0.25	0.17
L-Met	0.24	0.21	0.18	0.23	0.20	0.18	0.24	0.22	0.20
Total	100.00	100.00	100.00	100.00	100.00	100.00	100.00	100.00	100.00
Analysis results of nutrient level ^2^									
DM, %	94.66	94.06	94.96	94.63	95.24	95.50	96.48	95.60	94.66
Gross energy, J/kg	11.27	11.20	11.20	11.70	11.70	11.70	12.12	12.13	12.13
CP, %	13.53	15.40	17.26	14.80	15.33	16.71	14.44	15.35	16.93
Nutrient levels ^3^									
CP, %	14.00	15.52	17.01	14.00	15.50	17.03	14.04	15.50	17.01
ME, MJ/kg	11.28	11.28	11.29	11.70	11.70	11.69	12.12	12.12	12.13
Ca, %	1.07	1.07	1.07	1.07	1.07	1.07	1.07	1.07	1.07
P, %	0.75	0.75	0.75	0.75	0.75	0.75	0.75	0.75	0.75
AP, %	0.35	0.35	0.37	0.37	0.39	0.39	0.39	0.39	0.40
Ca/P	1.42	1.44	1.44	1.42	1.43	1.43	1.43	1.43	1.43
Lys, %	0.84	0.84	0.84	0.84	0.85	0.84	0.84	0.84	0.84
Met, %	0.41	0.41	0.41	0.41	0.40	0.41	0.41	0.41	0.41

Abbreviations: Low protein (LP), medium protein (MP), high protein (HP), low metabolizable energy (LE), medium metabolizable energy (ME), high metabolizable energy (HE). ^1^ The premix provided the following per kilogram of diets: vitamins: A 10,000 IU, D_3_ 3125 IU, E 2.5 mg, K_3_ 2.5 mg, B_1_ 2.5 mg, B_2_ 8.75 mg, B_6_ 3.75 mg, B_12_ 0.015 mg, biotin 0.18 mg, folic acid 0.75 mg, nicotinamide 37.5 mg, pantothenic acid 12.5 mg. minerals: Fe 100 mg, Cu 8 mg, Mn 120 mg, I 1 mg, Se 0.3 mg. ^2^ Analyzed in triplicates. ^3^ The nutritional levels were calculated based on the measured values of feed ingredients, with partial data referenced from the China Feed Composition and Nutritional Value Table (26th Edition, 2015) [18].

**Table 2 animals-15-02387-t002:** Effects of dietary ME and CP levels on growth performance of male Jingyuan chickens aged 7–18 weeks.

ME, MJ/kg	CP, %	ADG/g	ADFI/g	FCR
11.28	14.0	21.52 ^d^	78.99 ^ab^	3.67 ^a^
	15.5	21.26 ^d^	78.68 ^abc^	3.70 ^a^
	17.0	21.52 ^d^	76.93 ^c^	3.58 ^ab^
11.70	14.0	21.66 ^d^	78.54 ^abc^	3.62 ^ab^
	15.5	24.02 ^a^	77.66 ^bc^	3.23 ^d^
	17.0	22.67 ^c^	77.00 ^c^	3.40 ^c^
12.12	14.0	21.58 ^d^	77.59 ^bc^	3.60 ^ab^
	15.5	22.35 ^c^	78.71 ^abc^	3.52 ^bc^
	17.0	23.34 ^b^	79.83 ^a^	3.42 ^c^
SEM		0.187	0.240	0.030
Main effect means				
ME	11.28	21.43 ^b^	78.20	3.65 ^a^
	11.70	22.78 ^a^	77.73	3.42 ^b^
	12.12	22.42 ^a^	78.71	3.51 ^b^
	*p*-value	<0.001	0.139	<0.001
CP	14.0	21.59 ^b^	78.37	3.63 ^a^
	15.5	22.54 ^a^	78.35	3.49 ^b^
	17.0	22.51 ^a^	77.92	3.47 ^b^
	*p*-value	<0.001	0.558	<0.001
ME*CP	*p*-value	<0.001	0.011	<0.001

Abbreviations: Crude protein (CP), Metabolizable energy (ME), Average daily gain (ADG), Average daily feed intake (ADFI), Feed conversion ratio (FCR), Megajoule (MJ). Values within the same column with no superscripts or identical superscripts indicate no significant difference (*p* > 0.05), while different lowercase superscripts indicate a significant difference (*p* < 0.05).

**Table 3 animals-15-02387-t003:** Effects of ME and CP levels on apparent nutrient metabolism of Jingyuan chickens aged 18 weeks.

ME, MJ/kg	CP, %	Dry Matter %	Gross Energy %	Crude Protein %
11.28	14.0	64.77	62.47	67.03 ^ab^
	15.5	64.50	63.04	67.29 ^a^
	17.0	64.15	65.60	64.63 ^c^
11.70	14.0	64.25	65.37	65.85 ^abc^
	15.5	65.01	67.21	66.51 ^ab^
	17.0	64.42	67.08	65.56 ^bc^
12.12	14.0	64.38	67.23	62.70 ^d^
	15.5	63.95	66.33	64.67 ^c^
	17.0	64.19	66.66	64.32 ^c^
SEM		0.089	0.382	0.309
Main effect means				
ME	11.28	64.47	63.70 ^b^	66.32 ^a^
	11.70	64.56	66.55 ^a^	65.97 ^a^
	12.12	64.17	66.74 ^a^	63.90 ^b^
	*p*-value	0.135	<0.001	<0.001
CP	14.0	64.46	65.02	65.19 ^b^
	15.5	64.49	65.53	66.16 ^a^
	17.0	64.25	66.45	64.84 ^b^
	*p*-value	0.418	0.061	0.018
ME*CP	*p*-value	0.100	0.071	0.021

Values within the same column with no superscripts or identical superscripts indicate no significant difference (*p* > 0.05), while different lowercase superscripts indicate a significant difference (*p* < 0.05).

**Table 4 animals-15-02387-t004:** Effects of ME and CP levels on digestive organ index of Jingyuan chickens aged 18 weeks.

ME, MJ/kg	CP, %	Proventriculus/g	Gizzard/g	Duodenum/cm	Jejunoileum/cm	Cecum/cm
11.28	14.0	5.37	39.30 ^ab^	22.20 ^b^	107.75	36.00
	15.5	5.33	36.85 ^bc^	21.00 ^b^	113.50	38.00
	17.0	5.20	33.39 ^c^	23.20 ^b^	114.20	39.00
11.70	14.0	4.98	35.96 ^bc^	22.23 ^b^	97.50	32.60
	15.5	5.91	38.35 ^ab^	27.20 ^a^	121.25	33.80
	17.0	5.52	34.34 ^c^	21.50 ^b^	112.40	35.40
12.12	14.0	5.57	39.05 ^ab^	22.20 ^b^	108.20	36.20
	15.5	6.02	39.64 ^ab^	21.80 ^b^	121.00	38.00
	17.0	6.53	40.96 ^a^	26.25 ^a^	119.20	42.75
SEM		0.114	0.521	0.386	1.746	0.643
Main effect means						
ME	11.28	5.30 ^b^	36.54 ^b^	22.21 ^b^	112.00	37.54 ^a^
	11.70	5.38 ^b^	36.07 ^b^	23.80 ^a^	110.54	33.93 ^b^
	12.12	6.00 ^a^	39.77 ^a^	23.21 ^ab^	115.79	38.77 ^a^
	*p*-value	0.011	0.002	0.039	0.283	<0.001
CP	14.0	5.30	38.10	22.21	104.48 ^b^	34.93 ^b^
	15.5	5.75	38.28	23.50	118.58 ^a^	36.60 ^ab^
	17.0	5.75	36.23	23.62	115.27 ^a^	38.77 ^a^
	*p*-value	0.100	0.099	0.055	0.002	0.011
ME*CP	*p*-value	0.223	0.034	<0.001	0.471	0.699

Values within the same column with no superscripts or identical superscripts indicate no significant difference (*p* > 0.05), while different lowercase superscripts indicate a significant difference (*p* < 0.05).

**Table 5 animals-15-02387-t005:** Effects of ME and CP levels on intestinal morphology of Jingyuan chickens aged 18 weeks.

ME, MJ/kg	CP, %	Crypt Depth, CD/μm	Villus Height, VH/μm	VH/CD	Muscle Layer Thickness, MLT/μm
11.28	14.0	128.19 ^de^	702.44 ^c^	5.33 ^bc^	271.45 ^ab^
	15.5	131.10 ^cd^	699.45 ^c^	5.08 ^cd^	276.10 ^ab^
	17.0	135.15 ^abc^	628.31 ^d^	4.62 ^de^	279.55 ^a^
11.70	14.0	132.51 ^bcd^	744.55 ^b^	4.98 ^cd^	182.95 ^d^
	15.5	131.83 ^bcd^	799.02 ^a^	6.24 ^a^	280.55 ^a^
	17.0	130.32 ^cd^	684.28 ^c^	5.15 ^bc^	246.30 ^c^
12.12	14.0	139.53 ^a^	679.58 ^c^	5.11 ^bcd^	273.67 ^ab^
	15.5	123.68 ^e^	629.25 ^d^	4.39 ^e^	247.85 ^c^
	17.0	136.84 ^ab^	753.82 ^b^	5.60 ^b^	256.47 ^bc^
SEM		0.626	4.854	0.063	3.050
Main effect means					
ME	11.28	132.25	675.89 ^b^	4.97 ^b^	275.79 ^a^
	11.70	131.41	737.09 ^a^	5.44 ^a^	220.62 ^c^
	12.12	134.05	688.12 ^b^	5.12 ^b^	261.28 ^b^
	*p*-value	0.359	<0.001	<0.001	<0.001
CP	14.0	134.58 ^a^	703.63 ^ab^	5.12	227.78 ^b^
	15.5	129.38 ^b^	714.03 ^a^	5.33	270.05 ^a^
	17.0	133.65 ^a^	681.29 ^b^	5.10	259.74 ^a^
	*p*-value	<0.001	0.043	0.624	<0.001
ME*CP	*p*-value	<0.001	<0.001	<0.001	<0.001

Values within the same column with no superscripts or identical superscripts indicate no significant difference (*p* > 0.05), while different lowercase superscripts indicate a significant difference (*p* < 0.05).

## Data Availability

All data generated or analyzed during this study are included in this published article.

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
