# Peer review of "Effects of Dietary Metabolizable Energy and Crude Protein Levels on the Nutrient Metabolism, Gut Development and Microbiota Composition in Jingyuan Chicken"

_animals, 2025, doi:10.3390/ani15162387_

Round 1
Reviewer 1 Report
Comments and Suggestions for Authors
The aim of this study was to investigate the effects of diets with varying energy and protein levels on the nutrient metabolism, digestive organ traits, intestinal morphology, and microbial structure of male Jingyuan chickens aged 7–18 weeks. The introduction chapter contains a review of the world literature on the subject of the article. The research methods used are correct. The Materials and Methods and Results chapters require additions or improve in some places. The discussion is exhaustively correct. The Reference and Author Contributions chapters must be prepared in accordance with the requirements of Animals journal.
General comment
In the Material and Methods chapter there is no information about the type of building (closed, without enclosures), cage dimensions, floor type, about air humidity, type of light, color, intensity
The Reference chapter must be prepared in accordance with the Animals journal requirements: a full list of co-authors of individual items, initials first name, abbreviated name of the journal according to ISO4 in italic, year in bold, volume number in italic, page range long dash from the insert function instead of a short dash from the keyboard
Author Contributions, first action (only one), then authors, action 2: authors, etc.
For significance, use a low letter "p" in italic instead of "P" in the main article
Affiliation designations as 1,2,3,4 instead of a, b, c, d
Please add "a dot" after all table title
Detailed comments:
L1 "Article" instead of current form
L20, 26 microbiota or microbial?
L23 Abstract: The effects of...
L24-37 not bold for words in parentheses
L36 VH/CD ratio also significant
L62 and others – no spaces between reference numbers, for example [3,4] instead of [3, 4]
L106 please provide cage dimensions and floor type
L115-116 (NY/T 33-2004) Is this a reference?
L118 Please add a dot at the end of the table title.
In Tables 1, - 1, 2, and 3 for Premix, analysis results of nutrient levels, Nutrient levels as a superscript instead of the current form:
L119 Please add explanations for LE, ME, and HE.
L121 …..diets: vitamins – A 10,000 IU, D3 3,125 IU, etc. minerals:…
L127 (26th Edition, 2015) – Please provide a Reference.
L134 25 °C, space after the number.
L141 FCR is more commonly used in poultry practice and literature. Please change F/G to FCR.
L154-155 GB/T 6435-2014, GB/T 45104-2024, GB/T 6432-1994 – Please add a Reference for the standards.
L162 male chickens
L195 (p < 0.001) instead of (P < 0.01)
L197 for F/G CP17% is the lowest.
In Table 2, please add data for moisture % if available.
L203 Please add explanations for the abbreviations CP, ADG, ADFI, ME, MJ, FCR.
L214 14.00% or 15.50%?
In Tables 2-5, delete lines within the table.
L229 Delete "and duodenum length."
L245 for ME?
In Table 5, VH/CD ratio instead of VH/CD.
L254 No mention of Figure 1 or its brief description.
In Discussion chapter, add a space before the reference number.
L411 „traits” instead of development.
Author Response
We sincerely appreciate the reviewers' valuable comments and suggestions, which have greatly helped improve the quality of our manuscript. Below, we provide a point-by-point response to each comment.
General comments
Comments 1: In the Material and Methods chapter there is no information about the type of building (closed, without enclosures), cage dimensions, floor type, about air humidity, type of light, color, intensity
Response 1: Thank you for pointing this out. I agree with this comment. Accordingly, we have revised the text with the modifications highlighted in red.
Modified sentence(L136–142): “The experiment was conducted in a closed, environmentally controlled poultry house. A three-tier semi-stepped cage system was used, with two chickens per cage and even distribution across tiers. Each cage dimensions were 40 cm × 50 cm × 50 cm in width, length, and height. The environmental conditions were controlled at 20–25 °C, 50–60% relative humidity, and 16 h of light per day at 20 lux. Chickens underwent a 1-week adaptation period before the 10-week formal trial. Daily management included ventilation, cleaning, and routine health checks.”
Comments 2: The Reference chapter must be prepared in accordance with the Animals journal requirements: a full list of co-authors of individual items, initials first name, abbreviated name of the journal according to ISO4 in italic, year in bold, volume number in italic, page range long dash from the insert function instead of a short dash from the keyboard
Response 2: Thank you for pointing this out. I agree with this comment. I have revised the references in detail according to the Animals citation format.
Comments 3: Author Contributions, first action (only one), then authors, action 2: authors, etc.
Response 3: Thank you for pointing this out. I agree with this comment. Accordingly, we have revised the text with the modifications highlighted in red.
Modified text(L448–452): Author Contributions: Conceptualization, J. Liu and W.Z. Yang; data curation, J. Liu; formal analysis, Q.X. Gao; funding acquisition, G.S. Xin; methodology, X. Guo, J. Liu, and J. Yang; project administration, J. Zhang; supervision, J. Zhang and G.S. Xin; writing—original draft, X. Guo; writing—review and editing, X. Guo and G.S. Xin. All authors have read and agreed to the published version of the manuscript.
Comments 4: For significance, use a low letter "p" in italic instead of "P" in the main article
Response 4: Thank you for pointing this out. I agree with this comment. Accordingly, we have revised the text with the modifications highlighted in red.
Comments 5: Affiliation designations as 1,2,3,4 instead of a, b, c, d
Response 5: Thank you for pointing this out. I agree with this comment. Accordingly, we have revised the text with the modifications highlighted in red.
Comments 6: Please add "a dot" after all table title
Response 6: Thank you for pointing this out. I agree with this comment. Accordingly, we have revised the text with the modifications highlighted in red.
Detailed comments
Comments 1: L1 "Article" instead of current form
Response 1: Thank you for pointing this out. I agree with this comment. Accordingly, we have revised the text with the modifications highlighted in red.
Comments 2: L20, 26 microbiota or microbial?
Response 2: Thank you for pointing this out. I agree with this comment. We have changed “microbial composition” to “microbiota composition” and highlighted the revision in red.
Comments 3: L23 Abstract: The effects of...
Response 3: Thank you for pointing this out. I agree with this comment. Accordingly, we have revised the text with the modifications highlighted in red.
Comments 4: L24-37 not bold for words in parentheses
Response 4: Thank you for pointing this out. I agree with this comment. The manuscript has been revised according to the reviewers’ comments.
Comments 5: L36 VH/CD ratio also significant
Response 5: Thank you for pointing this out. I agree with this comment. Accordingly, we have revised the text with the modifications highlighted in red.
Original: “Moreover, this group exhibited significantly higher apparent metabolizable rates of CP, gizzard weight, duodenal length, jejunal VH, CD and MLT compared with those in the other groups (P <0.05).”
Modified L37-39: “Moreover, this group showed significantly higher duodenal length, jejunal CD, VH/CD and MLT compared with the other groups (p <0.05).”
Comments 6: L62 and others – no spaces between reference numbers, for example [3,4] instead of [3, 4]
Response 6: Thank you for pointing this out. I agree with this comment. The manuscript has been revised according to the reviewers’ comments.
Comments 7: L106 please provide cage dimensions and floor type
Response 7: Thank you for pointing this out. I agree with this comment. Accordingly, we have revised the text with the modifications highlighted in red. (Same as General Comments 1 above.)
Comments 8: L115-116 (NY/T 33-2004) Is this a reference?
Response 8: Thank you for pointing this out. I agree with this comment. Accordingly, we have revised the text with the modifications highlighted in red.
Added reference:
- China, Ministry of Agriculture of the People's Republic of China. Chicken Breeding Standard (NY/T33-2004). Hunan Feed 2006, 19–27.
Comments 9: L118 Please add a dot at the end of the table title.
Response 9: Thank you for pointing this out. I agree with this comment. Accordingly, we have revised the text with the modifications highlighted in red.
Comments 10: In Tables 1, - 1, 2, and 3 for Premix, analysis results of nutrient levels, Nutrient levels as a superscript instead of the current form:
Response 10: Thank you for pointing this out. I agree with this comment. Accordingly, we have revised the text with the modifications highlighted in red.
Comments 11: L119 Please add explanations for LE, ME, and HE.
Response 11: Thank you for pointing this out. I agree with this comment. Accordingly, we have revised the text with the modifications highlighted in red.
Original: low energy (LE), medium energy (ME), high energy (HE)
Modified L124–125: low metabolizable energy (LE), medium metabolizable energy (ME), high metabolizable energy (HE)
Comments 12: L121 …..diets: vitamins – A 10,000 IU, D3 3,125 IU, etc. minerals:…
Response 12: Thank you for pointing this out. I agree with this comment. Accordingly, we have revised the text with the modifications highlighted in red.
Original: The premix provided the following per kilogram of diet: VA 10,000 IU, VD3 3,125 IU, VE 2.5 mg, VK3 2.5 mg, VB1 2.5 mg, VB2 8.75 mg, VB6 3.75 mg, VB12 0.015 mg, D-biotin 0.18 mg, folic acid 0.75 mg, nicotinamide 37.5 mg, calcium pantothenate 12.5 mg, Fe 100 mg, Cu 8 mg, Mn 120 mg, I 1 mg, Se 0.3 mg.
Modified L126–130: The premix provided the following per kilogram of diets: vitamins: A 10,000 IU, D₃ 3,125 IU, E 2.5 mg, K₃ 2.5 mg, B₁ 2.5 mg, B₂ 8.75 mg, B₆ 3.75 mg, B₁₂ 0.015 mg, biotin 0.18 mg, folic acid 0.75 mg, nicotinamide 37.5 mg, pantothenic acid 12.5 mg. minerals: Fe 100 mg, Cu 8 mg, Mn 120 mg, I 1 mg, Se 0.3 mg.
Comments 13: L127 (26th Edition, 2015) – Please provide a Reference.
Response 13: Thank you for pointing this out. I agree with this comment. Accordingly, we have revised the text with the modifications highlighted in red.
Added reference:
- China, Feed Database in China, Tables of Feed Composition and Nutritive Values in China. 2015. http://www.chinafeeddata.org.cn
Comments 14: L134 25 °C, space after the number.
Response 14: Thank you for pointing this out. I agree with this comment. Accordingly, we have revised the text.
Comments 15: L141 FCR is more commonly used in poultry practice and literature. Please change F/G to FCR.
Response 15: Thank you for pointing this out. I agree with this comment. Accordingly, we have revised the text with the modifications highlighted in red.
Comments 16: L154-155 GB/T 6435-2014, GB/T 45104-2024, GB/T 6432-1994 – Please add a Reference for the standards.
Response 16: Thank you for pointing this out. I agree with this comment. Accordingly, we have revised the text with the modifications highlighted in red.
Added references:
- GB/T 6435-2014; Determination of moisture in feeds. China National Standard, Supervision: Beijing, China, 2014.
- GB/T 45104-2024; Determination of gross energy in feeds—Bomb calorimetry. China National Standard, Supervision: Beijing, China, 2024.
- GB/T 6432-1994; Determination of crude protein in feeds—Kjeldahl method. China National Standard, Supervision: Beijing, China, 1994.
Comments 17: L162 male chickens
Response 17: Thank you for pointing this out. I agree with this comment. Accordingly, we have revised the text with the modifications highlighted in red.
Comments 18: L195 (p < 0.001) instead of (P < 0.01)
Response 18: Thank you for pointing this out. I agree with this comment. Accordingly, we have revised the text with the modifications highlighted in red.
Comments 19: L197 for F/G CP17% is the lowest. In Table 2, please add data for moisture % if available.
Response 19: Thank you for pointing this out. I agree with this comment.
Original: The combination of 11.70 MJ/kg ME and 15.5% CP resulted in the highest ADG and the lowest F/G among the groups (p <0.05).
At the dietary ME level of 11.70 MJ/kg, the feed conversion ratio (FCR) values for the 14%, 15.5%, and 17% CP groups were3.62ab、3.23d、3.40c, respectively, with the 15.5% CP group showing the lowest FCR.
Supplementary text added (L212–213): At dietary ME levels of 11.70 or 12.12 MJ/kg, the 17.0% CP group showed a significantly lower FCR than the 14.0% CP groups (p <0.05).
About add data for moisture. Unfortunately, we were unable to collect moisture data during this experiment, although we ensured that the environmental humidity remained between 50–60%. Thank you for your valuable suggestion, and we will pay attention to this indicator in future relevant experiments.
Comments 20: L203 Please add explanations for the abbreviations CP, ADG, ADFI, ME, MJ, FCR.
Response 20: Thank you for pointing this out. I agree with this comment. Accordingly, we have revised the text with the modifications highlighted in red.
Supplementary text added (L218–219): crude protein (CP), Metabolizable energy (ME), Average daily gain (ADG), Average daily feed intake (ADFI), Feed conversion ratio (FCR), Megajoule (MJ).
Comments 21: L214 14.00% or 15.50%?
Response 21: Thank you for pointing this out. Upon verification, the correct value is 14.00%. Accordingly, we have revised the text with the modifications highlighted in red. (L228)
Comments 22: In Tables 2–5, delete lines within the table.
Response 22: Thank you for pointing this out. I agree with this comment. The lines within the tables have been removed.
Comments 23: L229 Delete "and duodenum length."
Response 23: Thank you for pointing this out.
When CP was 17%, and ME levels were 11.28, 11.70, and 12.12 MJ/kg respectively, the duodenum lengths were 23.20b, 21.50b, and 26.25a.
Specifically, when the CP level was 17%, the group of chickens receiving 12.12 MJ/kg ME exhibited significantly higher gizzard weights and duodenum length compared with chickens in the other ME groups (p < 0.05).
Comments 24: L245 for ME?
Response 24: Thank you for pointing this out. I agree with this comment. Accordingly, we have revised the text with the modifications highlighted in red.
Original: however, the CD was not significantly different between groups (P >0.05).
Modified L259: however, CD did not differ significantly among the CP groups (p > 0.05).
Comments 25: In Table 5, VH/CD ratio instead of VH/CD.
Response 25: Thank you for pointing this out. I agree with this comment. Accordingly, we have revised the text with the modifications highlighted in red.
Comments 26: L254 No mention of Figure 1 or its brief description.
Response 26: Thank you for pointing this out. I agree with this comment. Accordingly, we have revised the text with the modifications highlighted in red.
Original: Specifically, when the CP level was 15.5%, the group with 12.12 MJ/kg ME exhibited significantly lower CD, VH, and MLT values (P <0.05) compared with those in the other ME groups. When the ME was 11.70 MJ/kg, the 15.5% CP group showed significantly higher VH, VH/CD, and MLT values (P <0.05) compared with those in the other CP groups.
Modified L253–258: Specifically, when the CP level was 15.5%, the group receiving 12.12 MJ/kg ME exhibited significantly lower CD, VH, VH/CD and MLT values (p < 0.05) compared with those in the other ME groups, along with shorter villi and thinner mucosa (Figure 1 HEMP). Conversely, when the ME level was 11.70 MJ/kg, the 15.5% CP group showed significantly higher VH, VH/CD, and MLT values (p < 0.05) compared with those in the other CP groups, and the jejunal villi appeared more elongated and closely arranged (Figure 1 MEMP).
Comments 27: In Discussion chapter, add a space before the reference number.
Response 27: Thank you for pointing this out. I agree with this comment. We have revised the manuscript as requested.
Comments 28: L411 "traits" instead of "development."
Response 28: Thank you for pointing this out. I agree with this comment. Accordingly, we have revised the text with the modifications highlighted in red.

Reviewer 2 Report
Comments and Suggestions for Authors
This study systematically optimized dietary energy and protein levels and investigated their regulatory mechanisms governing growth for Jingyuan chicken. Overall, the experimental design is reasonable, the article content is detailed, and the topic selection has significant industrial significance. The specific suggestions for modification are as follows:
- Please adjust the use of "gastrointestinal development, and microbial composition" in the title. The "intestinal homeostasis" can be used as an alternative.
- The presentation format of the results of the two-factor analysis can be further optimized. It is suggested to adjust the presentation format of Table 2-5.
- It is best to unify the orientation of each intestinal section image in Figure 1.
- It is suggested that an association analysis be conducted between the microbiota and apparent nutrient metabolism or intestinal development.
The expression of the article can be further streamlined, especially for some indicators in the results that show no significant differences.
Author Response
We sincerely appreciate the reviewers' valuable comments and suggestions, which have greatly helped improve the quality of our manuscript. Below, we provide a point-by-point response to each comment.
Comment 1: Please adjust the use of "gastrointestinal development, and microbial composition" in the title. The "intestinal homeostasis" can be used as an alternative.
Response 1: Thank you for pointing this out. We fully agree with the suggestion and have revised the title accordingly, taking into account the reviewers’ valuable feedback:
“Effects of dietary metabolizable energy and crude protein levels on the nutrient metabolism, gut development, and microbiota composition in Jingyuan chicken.”
Comment 2: The presentation format of the results of the two-factor analysis can be further optimized. It is suggested to adjust the presentation format of Tables 2–5.
Response 2: Thank you for this helpful suggestion. We have revised the formatting of Tables 2–5 to enhance clarity and readability.
Comment 3: It is best to unify the orientation of each intestinal section image in Figure 1.
Response 3: Thank you for pointing this out. We have revised and replaced Figure 1 to ensure a consistent orientation of all intestinal section images.
Comment 4: It is suggested that an association analysis be conducted between the microbiota and apparent nutrient metabolism or intestinal development.
Response 4: Thank you for the suggestion. We conducted a correlation analysis between the microbiota and apparent nutrient metabolism and intestinal development. However, no statistically significant associations were found, so we did not include these results in the manuscript.
Comment 5: The expression of the article can be further streamlined, especially for some indicators in the results that show no significant differences.
Response 5: Thank you for the suggestion. We have revised and streamlined the relevant sections accordingly, with the changes highlighted in red in the revised manuscript.
Original (L32–40):
The levels of ME and CP, along with their interactions, had a significant effect on the average daily gain (ADG), average daily feed intake, feed-to-gain ratio (F/G), apparent metabolizable rate of CP, gizzard weight, duodenal and cecal lengths, jejunal villus height (VH), crypt depth (CD), and muscle layer thickness (MLT) (P<0.05). The combination of medium level ME (11.70 MJ/kg) and medium level CP (15.50%) (MEMP group) exhibited the best performance and exhibited the highest ADG and the lowest F/G (P <0.05). Moreover, this group exhibited significantly higher apparent metabolizable rates of CP, gizzard weight, duodenal length, jejunal VH, CD, and MLT compared with those in the other groups (P <0.05).
We have streamlined certain non-significant indicators. We deleted “cecal lengths,” “metabolizable rates of CP,” and “gizzard weight,” and replaced some repetitive terms such as “exhibited.”
Revised (L31–39):
The levels of ME and CP, along with their interactions, had significant effects on the average daily gain (ADG), average daily feed intake, feed conversion ratio (FCR), apparent metabolizable rate of CP, gizzard weight, duodenal lengths, jejunal villus height (VH), crypt depth (CD), and muscle layer thickness (MLT) (p <0.05). The combination of medium level ME (11.70 MJ/kg) and medium level CP (15.50%) (MEMP group) exhibited the best performance, with the highest ADG and the lowest FCR (p <0.05). Moreover, this group showed significantly higher duodenal length, jejunal CD, VH/CD and MLT compared with the other groups (p <0.05).
Original (L208):
Different levels of ME and CP significantly affected the apparent metabolic rates of GE and CP (P <0.01).
The descriptions of the results have been further revised and clarified for better readability and comprehension.
Revised (L223–226):
Different levels of ME and CP significantly affected the apparent metabolic rates of CP, with a notable interaction between the two factors (p <0.05). Dietary ME level significantly influenced the apparent metabolizable energy of GE, while CP level had no noticeable impact (p >0.05).
Original (L226–230):
Specifically, when the CP level was 15.5%, the group with 12.12 MJ/kg ME exhibited significantly lower CD, VH, and MLT values (P <0.05) compared with those in the other ME groups. When the ME was 11.70 MJ/kg, the 15.5% CP group showed significantly higher VH, VH/CD, and MLT values (P <0.05) compared with those in the other CP groups.
We have supplemented the description of Figure 1.
Revised (L253–258):
Specifically, when the CP level was 15.5%, the group receiving 12.12 MJ/kg ME exhibited significantly lower CD, VH, VH/CD and MLT values (p < 0.05) compared with those in the other ME groups, along with shorter villi and thinner mucosa (Figure 1 HEMP). Conversely, when the ME level was 11.70 MJ/kg, the 15.5% CP group showed significantly higher VH, VH/CD, and MLT values (p < 0.05) compared with those in the other CP groups, and the jejunal villi appeared more elongated and closely arranged (Figure 1 MEMP).
Reviewer 3 Report
Comments and Suggestions for Authors
The present study investigated the effects of dietary metabolizable energy and crude protein levels on the apparent nutrient metabolism, gastrointestinal development, and microbial composition in Jingyuan chicken. The results showed that the combination of medium level ME and CP was the most favorable. Following are some comments on this manuscript.
Title: The title is a bit too long. “apparent” could be deleted. “gastrointestinal” could be replaced with “gut”.
L76-88: Indigenous breeds are different from the commercial breeds with slower growth performance. The authors should further introduce why 7-18 wk was selected in this study. Is this a rapid growth period for Jingyuan chicken?
L107-110: What was the main criterion for choosing these doses of ME and CP? This should be involved in the text with related references.
Materials and Methods: It is suggested to add a section of sample collection separately.
L177-178: Why did the authors choose these 5 groups of total 9 groups for 16s sequencing? More explanations should be provided here.
L321: The authors should compare the current data (effects of ME and CP levels on the growth performance and apparent nutrient metabolism) with other indigenous chickens.
Discussion: The limitations of this study should be discussed in the text. In addition, the authors are suggested to add a paragraph about the implications of this study on the indigenous chicken industry in the end of discussion.
Author Response
We sincerely appreciate the reviewers' valuable comments and suggestions, which have greatly helped improve the quality of our manuscript. Below, we provide a point-by-point response to each comment.
Comments 1: Title: The title is a bit too long. “apparent” could be deleted. “gastrointestinal” could be replaced with “gut”.
Response 1: Thank you for pointing this out. I agree with this comment. Based on the reviewers’ suggestions, we have revised the title to: “Effects of dietary metabolizable energy and crude protein levels on the nutrient metabolism, gut development, and microbiota composition in Jingyuan chicken”
Comments 2: L76-88: Indigenous breeds are different from the commercial breeds with slower growth performance. The authors should further introduce why 7-18 wk was selected in this study. Is this a rapid growth period for Jingyuan chicken?
Response 2: Thank you for pointing this out. I agree with this comment. Accordingly, we have revised the text with the modifications highlighted in red.
Supplementary text added (L84–90): Notably, under the local rearing system, Jingyuan chickens are raised in a centralized facility during the brooding phase and then transferred to individual farmers for the growing period from 7 to 18 weeks of age. Despite the importance of this phase for rapid growth and gastrointestinal development, a standardized dietary formulation tailored to this specific stage is still lacking. At 18 weeks of age, Jingyuan chickens reach the onset of sexual maturity, by which time the digestive system has generally completed its structural and functional development [15].
- de Los Mozos, J.; García-Ruiz, A.I.; den Hartog, L.A.; Villamide, M.J. Growth curve and diet density affect eating motivation, behavior, and body composition of broiler breeders during rearing. Poult. Sci. 2017, 96, 2708–2717. https://doi.org/10.3382/ps/pex045
Comments 3: L107-110: What was the main criterion for choosing these doses of ME and CP? This should be involved in the text with related references.
Response 3: Thank you for pointing this out. I agree with this comment. Accordingly, we have revised the text with the modifications highlighted in red.
Modified sentence (L119–121): ME and CP levels were primarily based on China’s Feeding Standard of Chickens (NY/T 33—2004) [17], with appropriate adjustments made to the ME and CP settings.
- China, Ministry of Agriculture of the People's Republic of China. Chicken Breeding Standard (NY/T33-2004). Hunan Feed 2006, 19–27.
Comments 4: Materials and Methods: It is suggested to add a section of sample collection separately.
Response 4: Thank you for pointing this out. I agree with this comment. Accordingly, we have revised the text with the modifications highlighted in red.
Supplementary text added (L143–168):
2.3. Sample Collection
The initial and final weights of Jingyuan chickens were recorded at the commencement and conclusion of the formal trial, respectively. During the trial, the feed intake was recorded on a weekly basis, and the average daily feed intake (ADFI), average daily gain (ADG), and feed conversion ratio (FCR) were calculated. The number of deceased Jingyuan chickens was recorded on a daily basis, and the FCR was corrected based on the mortality data.
One week before the end of the trial, a metabolism experiment was conducted using the total fecal-collection method. Feed intake and excreta weight were accurately recorded for each replicate over 1 week, and contemporaneous feed samples were collected for subsequent nutrient analysis. Excreta samples collected in trays were inspected daily. Foreign matter (e.g., feathers) was removed and the samples were sprayed with an appropriate amount of 10% concentrated sulfuric acid for nitrogen fixation. The weights of fresh excreta samples were recorded. Subsequently, fresh excreta and feed samples were dried, weighed, ground, sieved, and stored for subsequent chemical analysis [19].
After 12 h of fasting at the end of the formal trial, six male chickens from each group with body weights close to the average body weight of the group were randomly selected for slaughter. Chickens were anesthetized by inhalation of ether until loss of consciousness, followed by humane euthanasia via severing of the carotid artery. The glandular stomach and gizzard were emptied of their contents and weighed, and the lengths of the duodenum, jejunoileum, and cecum were measured [20].
A 3–5-cm segment of jejunal tissue (located 5–10 cm below the junction of the duodenum and the jejunum) was excised, gently flushed with physiological saline to remove chyme, and fixed in 4% formaldehyde solution for 24 hours for morphological analysis. cecal contents were collected and snap-frozen in liquid nitrogen for microbiota sequencing.
Comments 5: L177-178: Why did the authors choose these 5 groups of total 9 groups for 16s sequencing? More explanations should be provided here.
Response 5: Thank you for pointing this out. I agree with this comment. Accordingly, we have revised the text with the modifications highlighted in red.
Supplementary text added (L187–191): Based on growth performance and intestinal development, the MEMP group exhibited the best results. To compare the effects of different energy levels at a medium protein level, the LEMP, MEMP, and HEMP groups were selected. Similarly, to assess the impact of varying protein levels at a medium energy level, the MELP, MEMP, and MEHP groups were chosen.
Comments 6: L321: The authors should compare the current data (effects of ME and CP levels on the growth performance and apparent nutrient metabolism) with other indigenous chickens.
Response 6: Thank you for pointing this out. I agree with this comment. Accordingly, we have revised the text with the modifications highlighted in red.
Supplementary text added (L338–340): The results of this study demonstrate that dietary metabolizable energy (ME) and crude protein (CP) levels have a significant interactive effect on the growth performance of Jingyuan chickens, consistent with the observations of Zhu et al [26].
Supplementary text added (L346–357): The study found that feed intake decreased significantly in Jingyuan chickens with increasing CP levels, while elevated ME levels promoted feed intake. Ko et al. reported that feed intake significantly decreased in male Cobb broilers when the protein level was 19% and dietary energy decreased by 100 kcal/kg, indicating a correlation between dietary energy level and feed intake [3]. However, previous studies have indicated that feed intake and FCR of Beijing-You chickens and Taihe silky fowls were significantly affected by dietary ME levels, and increasing ME levels would lead to reductions in both feed intake and FCR in these breeds [7,9]. In addition, this study observed a significant interaction between ME and CP levels on Jingyuan chicken FCR. In contrast, increasing dietary CP resulted in higher FCR in male Arbor Acres broilers, while ME levels had no significant effect [5]. This demonstrates breed-specific differences in the nutritional responses to dietary energy and protein.
Comments 7: Discussion: The limitations of this study should be discussed in the text. In addition, the authors are suggested to add a paragraph about the implications of this study on the indigenous chicken industry in the end of discussion.
Response 7: Thank you for pointing this out. I agree with this comment. Accordingly, we have revised the text with the modifications highlighted in red.
Supplementary text added (L364–373): However, this study has certain limitations. The dietary recommendation derived from the MEMP group is applicable only to male Jingyuan chickens, and its relevance to females remains untested. In addition, the experimental period was limited to 7–18 weeks, without evaluating the early brooding phase or other production stages. Further research covering other developmental periods is needed to support recommendations for the entire Jingyuan chicken production. Nevertheless, the findings are valuable for the indigenous Jingyuan chicken industry. By identifying the optimal combination of dietary energy and protein levels, this study contributes to the development of tailored feeding strategies for Jingyuan chickens, which can improve production efficiency and gut health, while also supporting the development of regional poultry farming.
Round 2
Reviewer 3 Report
Comments and Suggestions for Authors
The manuscript has been improved.